

# Pereskia bleo augments NK cell cytotoxicity against triple-negative breast cancer cells (MDA-MB-231)

Taif Kareem Khalaf[1], Norzila Ismail[1], Nor Amalia Nazri[1], Naveed Ahmed[2], Aidy Irman Yajid[3], Rohimah Mohamud[4] and Ramlah Kadir[4]

[1] Department of Pharmacology, School of Medical Sciences, Universiti Sains Malaysia, Kubang Kerian, Kelantan, Malaysia

[2] Department of Medical Microbiology and Parasitology, School of Medical Sciences, Universiti Sains Malaysia, Kubang Kerian, Kelantan, Malaysia

[3] Department of Pathology, School of Medical Sciences, Universiti Sains Malaysia, Kubang Kerian, Kelantan, Malaysia

[4] Department of Immunology, School of Medical Sciences, Universiti Sains Malaysia, Kubang Kerian, Kelantan, Malaysia

Corresponding author
Norzila Ismail,
norzila_ismail@usm.my

## ABSTRACT

**Background.** Natural killer cells (NK cells) are essential in cancer immunosurveillance in the body as they can recognize cancer cells that lacking MHC class 1 on their surface. Regulatory cytokines, including interleukin (IL)-18, IL-12, IL-10, IL-8, interferon-$\gamma$ (IFN-$\gamma$), and secretory granules like perforin and granzyme are involved in NK cell-mediated cytotoxicity. Stimulating NK cells cytotoxicity towards cancer cells is an ideal strategy to combat cancer naturally. Medicinal plants have been reported to enhance immunity, with *Pereskia bleo* (*P. bleo*) particularly noteworthy due to its abundant bioactive compounds and ability to activate immune cells. This study aimed to evaluate the potential of methanol extract of *P. bleo* leaves (MEPB) for enhancing NK cell cytotoxicity against triple-negative human breast cancer cells (MDA-MB-231).

**Methods.** The optimal concentration of MEPB to activate NK cells was determined using healthy blood samples, assessing the expression of IL-12, IL-18, IL-10, IL-8, IFN-$\gamma$, perforin, and granzyme B *via* an enzyme-linked immunosorbent assay (ELISA). NK cell purity from healthy donors and breast cancer patients was determined using specific antibodies, and the number of NK cells was assessed using flow cytometry and a hemocytometer. A co-culture experiment, ELISA, and apoptosis assay were used to evaluate NK-mediated cytotoxicity pathways.

**Results.** ELISA data indicated that MEPB at 7.5 μg/ml significantly increased the expression of IFN-$\gamma$, IL-12, IL-18, perforin, and granzyme B while decreasing IL-8 and IL-10 expression after 20 hrs of incubation. The average NK cell purity was 87.09 ± 0.043%. Breast cancer patients exhibited lower NK cell counts than healthy donors. Co-culture experiments demonstrated that NK cells induced apoptosis in MDA-MB-231 breast cancer cells in the presence of MEPB by increasing perforin, granzyme B, and IFN-$\gamma$ expression in both healthy donors and breast cancer patients-experimental groups. *P. bleo* enhances NK cell activation, promoting the apoptosis of triple-negative human breast cancer cells (MDA-MB-231), suggesting the potential use of MEPB leaves as an anti-cancer immunostimulant.

# INTRODUCTION

Cancer is a global health challenge, being the leading cause of mortality. In 2020, approximately 2.26 million new cases of breast cancer were diagnosed globally, constituting nearly 32% (8,418 new cases) of all malignancies among Malaysian women (*Mathan, Rajput & Singh, 2022*). Breast cancer classification, based on hormone receptor expression, delineates four subtypes: Oestrogen Receptor-Positive ($ER^+$), Progesterone Receptor-Positive ($PR^+$), Human Epidermal Growth Factor Receptor 2-Positive ($HER2^+$), and the notably aggressive triple-negative breast cancer (TNBC). TNBC, known as human epidermal growth factor receptor 2-negative ($HER2^-$), is a more aggressive breast cancer subtype than others and currently has limited treatment options available (*Alshaibi et al., 2022*).

While traditional cancer treatments, such as surgery, chemotherapy, and radiation therapy, persist, they still carry adverse effects (*Mathan, Rajput & Singh, 2022*). The emergence of immunotherapy has revolutionized cancer treatment, leveraging the body's immune system to target cancerous cells (*Baggio et al., 2017*). Among the diverse strategies encompassed by cancer immunotherapy, natural killer (NK) cells are one potential effector, exhibiting anti-cancer, and anti-microbial activities within the innate immune system (*Barrow et al., 2018*). In a healthy adult, NK cells ($CD3^-CD56^+$) constitute approximately 5–10% of lymphocytes. These cells have demonstrated cytotoxicity against various cancer cells while sparing normal cells through pathways, such as granule exocytosis (*e.g.*, perforin and granzymes), tumour necrosis factor (TNF)-related apoptosis-inducing ligand, Fas ligand, antibody-dependent cellular cytotoxicity (ADCC), and pro-inflammatory cytokines, including IFN-$\gamma$ (*Barrow et al., 2018*; *Chiossone et al., 2018*). A previous study reported that breast and prostate cancer patients have better results when their NK cells' activating receptors are increased (*Ascierto et al., 2013*). It is important to note that plant-derived bioactive compounds have been associated with increased NK cell cytotoxicity responses against target cells (*Barrow et al., 2018*; *Chiossone et al., 2018*). *In vivo* study investigated the effect of ashwagandha extracts (*Withania somnifera*) on healthy subjects' immune cells when given milk. After 96 h, the researchers found a considerable increase in NK cell activity, as evidenced by greater expression of CD69 (*Mikolai et al., 2009*).

Medicinal plants, rich in secondary metabolites with various biological activities (*Zebeaman et al., 2023*), have garnered attention for their potential to stimulate NK cells. For instance, *Sesamum indicum* extract has shown a significant increase in NK cytotoxicity against YAC-1 cancer cells. Aqueous ginseng extract has demonstrated an essential role in NK cell-mediated cytotoxicity through IFN-$\gamma$ activation, while thymoquinone isolation from *Nigella sativa* increased NK cell cytotoxicity against MCF-7 breast cancer cells by expressing perforin, granzyme B, and IFN-$\alpha$ (*Lucas et al., 2010*).

Radiation and chemotherapeutic medicines are used to treat cancer today, and both have significant side effects on patients. Thus, an increased global focus on finding nontoxic treatments for healthy cells that are toxic to cancer cells has emerged. *Pereskia bleo* (*P. bleo*) has been proven in earlier investigations to have anti-cancer properties. *P. bleo*, a traditional medicinal herb widely used throughout Asia, including Malaysia, has drawn attention due to its purported efficacy in treating various ailments, including cancer (*Lucas et al., 2010*). With an abundant profile of secondary metabolites, including alkaloids, fatty acids, glycosides, lactones, phenolic compounds, sterols, terpenoids, and carotenoids, *P. bleo* leaves have demonstrated cytotoxic effects against cancer cells in previous investigations (*Zebeaman et al., 2023*). To provide useful basic pharmacological information on this medicinal plant, further research is needed to understand the potential of *P. bleo* extract to produce cytotoxicity and increase immunological activation. Hence, this study aimed to investigate the ability of *P. bleo* leaves to stimulate NK cells and induce apoptosis in triple-negative human breast cancer cell lines (MDA-MB-231).

## MATERIALS AND METHODS

### Plant material

*P. bleo* leaves were sourced from Kota Bharu, Kelantan, Malaysia, authenticated, and a voucher specimen (USM 11730) was submitted to the Herbarium Unit, School of Biological Sciences, Universiti Sains Malaysia, Penang.

### Preparation of plant extract

Fresh *P. bleo* leaves were rinsed under running tap water and dried thoroughly in a hot-air oven at 50 °C for three days. The dried leaves were powdered using an electronic blender and extracted using a Soxhlet apparatus using hexane, ethyl acetate, and methanol successively. The experiment has shown that methanol extract has the highest cytotoxicity towards breast cancer cells. Therefore, we chose methanol extract for the current experiment. The extract was concentrated in a rotary evaporator, then weighed and diluted with dimethyl sulfoxide (DMSO) to prepare serial concentrations of 60, 30, 15, 7.5, and 3.75 µg/ml; these were kept at −20 °C for further use.

### Cell culture

MDA-MB-231 breast cancer cell lines were purchased from the American Type Culture Collection (ATCC, US) and cultured in Dulbecco's Modified Eagles Medium (DMEM (1X; Gibco™), while NK cells, isolated from healthy and breast cancer patient donors (women), were maintained in RPMI-1640 medium and kept in an incubator at 37 °C in a humidified incubator with 5% CO$_2$. Both mediums were supplemented with 5% fetal bovine serum (FBS; Gibco™, Waltham, MA, USA) and 1% penicillin-streptomycin (Gibco™, Waltham, MA, USA).

### Ethical approval

Ethical approval was obtained from JEPeM, Universiti Sains Malaysia, with the approval code: USM/JEPeM/20080430.

**Table 1  Blood sample experimental groups.**

| Groups | MEPB range of concentration | Incubations (µl) | | | | |
|---|---|---|---|---|---|---|
| | | MEPB (extract) | DMSO | RPMI-1640 medium | PHA | Blood |
| | 60 µg/ml +PHA | 10 | – | 190 | 100 | 700 |
| | 30 µg/ml +PHA | 10 | – | 190 | 100 | 700 |
| A | 15 µg/ml +PHA | 10 | – | 190 | 100 | 700 |
| | 7.5 µg/ml +PHA | 10 | – | 190 | 100 | 700 |
| | 3.75 µg/ml +PHA | 10 | – | 190 | 100 | 700 |
| B | PHA only | – | – | 200 | 100 | 700 |
| | 60 µg/ml | 10 | – | 290 | – | 700 |
| | 30 µg/ml | 10 | – | 290 | – | 700 |
| C | 15 µg/ml | 10 | – | 290 | – | 700 |
| | 7.5 µg/ml | 10 | – | 290 | – | 700 |
| | 3.75 µg/ml | 10 | – | 290 | – | 700 |
| D | DMSO only | – | 10 | 290 | – | 700 |

Notes.

Total volume = 1,000 µl.

Stimulator used: PHA for IFN-$\gamma$ and IL-8, IL-10, IL-12, IL-18, perforin, and granzyme B.

## Blood sample collection

Blood samples were collected from healthy Malaysian women ($n = 3$) and breast cancer patient donors ($n = 3$) aged 18–60 years. The donors were non-smokers, non-pregnant, not diagnosed with other diseases, and not undergoing any other treatment (either surgery, chemotherapy, or radiation). Additionally, breast cancer patient donors were selected from stages II-IV that did not receive chemotherapy. Written informed consent was obtained from each of the participants.

## Quantification of the optimal MEPB concentration

Healthy fresh blood donors (9 ml) were divided into four groups to quantify the optimal concentration of MEPB for subsequent analyses. Table 1 shows experimental groups that were created for the current investigation. Phytohaemagglutinin (PHA) (2 µg/mL) dissolved in sterile phosphate-buffered saline (PBS) was used as a cytokine stimulatory agent in the samples.

In order to assess the cytokines production (either by NK cells or other cells in blood) that contribute to NK cells activation, blood incubation protocols were carried out using methods by Ismail and co-workers with slight modification (*Ismail et al., 2018*). All blood samples were incubated for 20 hrs in 5% $CO_2$ at 37 °C. After incubation, blood samples were centrifuged at 1000 g for 15 min, and the supernatant was isolated to quantify the levels of IL-18, IL-12, IL-10, IL-8, IFN-$\gamma$, perforin, and granzyme by ELISA. Each well was filled with 100 µl of diluted standard, blank, and sample (supernatant). IFN-$\gamma$, IL-8, IL-10, IL-12, and IL-18 plates were incubated for 90 min at 37 °C, whereas perforin and granzyme B plates were incubated for 2 hrs at room temperature (RT). Then, the solution was discarded and washed with wash buffer (350 µl). Biotinylated detection Ab working solution (100 µl) was added to each well. Then, IFN-$\gamma$, IL-8, IL-10, IL-12, and IL-18 plates

were incubated for 60 min at 37 °C, while perforin and granzyme B plates were incubated for 1 h at RT. After incubation, the solution was discarded and washed five times with wash buffer (350 µl). Horseradish peroxidase (HPR) conjugate working solution (100 µl) was injected into each well. IFN-$\gamma$, IL-8, IL-10, IL-12, and IL-18 plates were incubated for 30 min at 37 °C, while perforin and granzyme B plates were incubated for 1 h at RT. Then, the solution from each well was discarded and washed five times with wash buffer (350 µl). In the dark, 90 µl of substrate reagent for IFN-$\gamma$, IL-8, IL-10, IL-12, and IL-18 plates and 100 µl for perforin and granzyme B plates were added to each well. IFN-$\gamma$, IL-8, IL-10, IL-12, and IL-18 plates were incubated for 15 min at 37 °C. Perforin and granzyme B plates were incubated at RT for 15 min. Finally, 50 µl of stop solution was added for IFN-$\gamma$, IL-8, IL-10, IL-12, and IL-18 plates and 100 µl for perforin and granzyme B plates. According to the manufacturer's instructions, the experiment was achieved using the Human ELISA Kit from Elabscience®, Fine Test®, and the ELISAPRO Kit (Mabtech, Nacka Strand, Sweden). An ELISA reader measured the absorbance (OD) data at 450 nm.

## Isolation of NK cells

A human NK Cell Isolation Kit (MACS, Miltenyi Biotec, Bergisch Gladbach, Germany) was used to isolate NK cells. The test was performed according to the manufacturer's protocol. Briefly, fresh blood was mixed with phosphate-buffered saline (PBS) (10:10 v/v), layered with 10 ml of lymphocyte separation medium, and centrifuged at 400 g for 30 min. After centrifugation, peripheral blood mononuclear cells (PBMCs) were collected, rinsed twice in 30 ml of PBS, and then resuspended in RPMI-1640 medium supplemented with 5% FBS and 1% penicillin-streptomycin. This mixture was incubated overnight (5% $CO_2$ at 37 °C) to allow the adherence of monocytes at the bottom of the flask. After incubation, the medium was collected and rinsed twice with PBS at 300 g for 10 min. The pellet was resuspended in 40 µl of NK buffer. The NK Cell Biotin Antibody Cocktail (10 µl) was added and incubated at 2–8 °C for 5 min. The mixture was incubated with 20 µl of NK Cell MicroBead Cocktail at 2–8 °C for 10 min. Following the incubation, the mixture was resuspended with 1,000 µl of NK buffer. The LS column was placed in the magnetic field of the MACS separator and rinsed with 3 ml of NK buffer before use. The cell suspension was added to the LS column and rinsed with 3 ml of NK buffer. The mixture was centrifuged at 300 g for 5 min. After being rinsed twice with PBS, 100 µl of cells were blocked with bovine serum albumin (BSA), incubated for 10 min at RT, and then rinsed twice with PBS at 300 g for 3 min. In complete darkness, 5 µl of PE-conjugated anti-CD56-PE and FITC-conjugated anti-CD3-FITC antibodies were added and incubated on ice for 30 min. The cells were rinsed with PBS at 300 g for 5 min and resuspended in 500 µl of PBS. The NK cells' purity and count number were evaluated using BD FACSCanto II flow cytometer System (two lasers, six channels/colors). Various techniques like antibody titration, compensation controls, and fluorescence minus one (FMO) were utilized to optimize that flow cytometry assay. The acquired data was analyzed using the FCS Express 7 software.

## NK cells cytotoxicity test

This experiment was performed to analyse the MEPB's ability to activate NK cells and induce apoptosis on MDA-MB-231 breast cancer cell lines. NK cells were isolated, and

**Table 2  NK cells experimental groups.**

| Groups | Incubation the co-culture | | Time incubation (hrs) | Concentration of MEPB leaves (μg/ml) |
|---|---|---|---|---|
| | **Donors** | | | |
| | **Healthy** | **Breast cancer patient** | | |
| Group 1 | MDA-MB-231 breast cancer cell lines alone | MDA-MB-231 breast cancer cell lines alone | | Negative control (Untreated) |
| Group 2 | MDA-MB-231 breast cancer cell lines + MEPB leaves | MDA-MB-231 breast cancer cell lines + MEPB leaves | | 7.5 μg/ml |
| Group 3 | MDA-MB-231 breast cancer cell lines + NK cells | MDA-MB-231 breast cancer cell lines + NK cells | 24 hrs | – |
| Group 4 | MDA-MB-231 breast cancer cell lines +NK cells + MEPB leaves | MDA-MB-231 breast cancer cell lines +NK cells + MEPB leaves | | 7.5 μg/ml |

**Notes.**

Final volume: 2 mL.

500 μl of NK cells ($2.15 \times 10^5$) and 200 μl of MDA-MB-231 cell lines ($10.75 \times 10^3$).

3 μl of MEPB leaves.

Complete the volume with RPMI-1640 medium.

MDA-MB-231 breast cancer cell lines were co-cultured at a 20:1 ratio (*Nishimura et al., 2017*). From the ELISA experiment, it was determined that 7.5 μg/ml of MEPB was the optimal concentration to be used for the co-cultured assay. Table 2 illustrated experimental groups that were created for this study.

All experiment groups were incubated at 37 °C with 5% $CO_2$ for 24 hrs. After incubation, the medium (supernatant) was collected and used to assess the level of IFN-$\gamma$, perforin, and granzyme B by ELISA. The study was performed according to the manufacturer's protocol (Elabscience® and ELISAPRO Kit (Mabtech, Nacka Strand, Sweden)). Subsequently, MDA-MB-231 breast cancer cell lines were subjected to the apoptosis assay (FITC Annexin V Apoptosis Detection Kit I) to determine cell death. The current experiment followed the manufacturer's instructions for the Annexin V-FITC/PI Kit. Briefly, MDA-MB-231 breast cancer cell lines ($10.75 \times 10^3$) were harvested using trypsinization (300 μl) and centrifuged at 300 g for 3 min. After being washed twice with cold PBS and re-suspended with 1 ml of 1X annexin-binding buffer dilution, 5 μl of Annexin V-FITC and 5 μl of propidium iodide were added to 100 μl of the re-suspended sample. The combination was incubated for 15 min at RT in complete darkness, followed by an addition of 400 μl of 1X annexin-binding buffer dilution at the experiment's conclusion. Instantaneous data analysis was performed using flow-cytometry and FCS7 Express cytometry software version 2019. The total number of experiments was carried out in triplicate to ensure accuracy.

## Statistical analysis

All the values were analysed with GraphPad Prism version 8.0.1. A one-way analysis of variance (ANOVA) was followed by Dunnett's *post-hoc* test to compare the data. *P* <0.05 refers to a significant value.

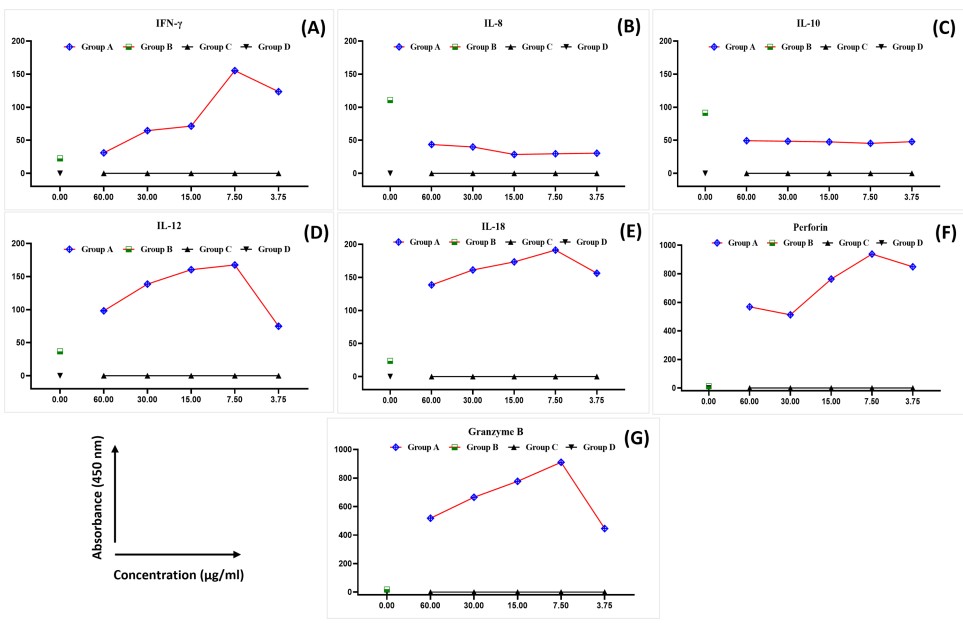

**Figure 1** **The expression of IFN-$\gamma$ (A), IL-8 (B), IL-10 (C), IL-12 (D), IL-18 (E), perforin (F), and granzyme B (G) in healthy blood samples.** The data was reported as mean $\pm$ SEM, $n = 3$.

## RESULTS

### Quantification of the optimal MEPB concentration

The levels of IFN-$\gamma$, IL-8, IL-10, IL-12, IL-8, perforin, and granzyme B were analysed in the blood samples collected from healthy female blood donors to determine the optimal MEPB concentration for the subsequent analyses. There was a substantial difference in the expression of cytokines between the examined groups (Fig. 1). The expression of IFN-$\gamma$, IL-12, IL-18, perforin, and granzyme B progressively rose in all serial concentrations of MEPB in Group A (+PHA + RPMI-1640 medium + MEPB) while dropping significantly in Group B (+PHA and +RPMI-1640 medium). In contrast, there was a drop in the expression of IL-10 and IL-8 in all serial concentrations of MEPB in Group A (+PHA + RPMI-1640 medium + MEPB), while it surged in Group B (+PHA + RPMI-1640 medium). The expression of IFN-$\gamma$, IL-12, IL-18, IL-10, IL-8, perforin, and granzyme B in Groups C (+RPMI-1640 medium + MEPB) and group D (+DMSO + RPMI-1640 medium) was equally unaffected. The data indicated that 7.5 µg/ml of MEPB was the optimal concentration for triggering IFN-$\gamma$, IL-12, IL-18, perforin, and granzyme B; it minimised the production of IL-10 and IL-8 in healthy blood samples (Fig. 1). Thus, 7.5 µg/ml was used for the co-cultured experiment. This finding suggests that MEPB plays a crucial role in promoting immunostimulants.

### Isolation and purification of NK cells data

Specific antibodies were used to determine the percentage of NK cell expression (anti-CD3-FITC and anti-CD56-PE, Santa Cruz Biotechnology, Dallas, TX, USA), as shown in Fig. 2. The current findings revealed highly pure NK cell isolation (87.09 $\pm$ 0.043%) from

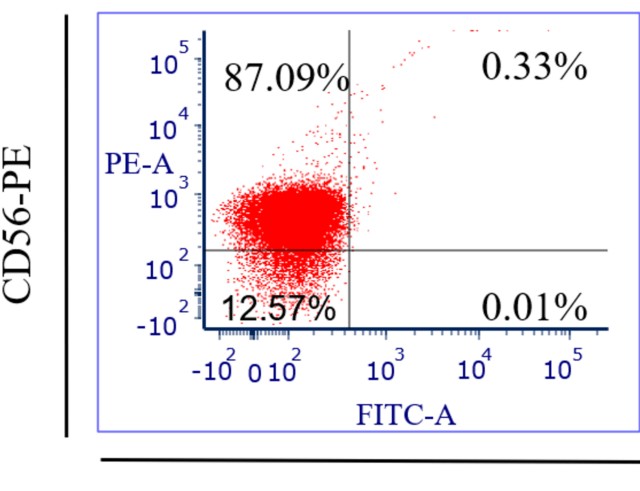

**Figure 2** **Flow-cytometry shows the percentage (%) of NK cells' purification isolated from healthy donors ($n = 3$) and breast cancer patients ($n = 3$) using a human isolation kit (Miltenyi Biotech).**

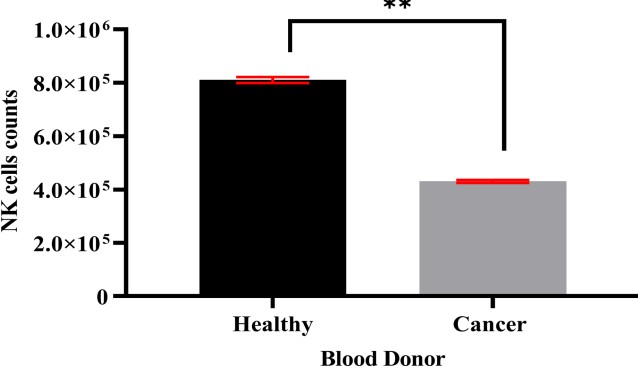

**Figure 3** **A count of NK cells isolated from healthy donors ($n = 3$) ($8.1 \times 10^5$ cells/ml) and breast cancer patients ($n = 3$) ($4.3 \times 10^5$ cells/ml).** The data is represented as mean $\pm$ SEM. $P < 0.05$ shows a significant difference between groups.

the donors and the breast cancer patients ($4.3 \times 10^5$ cells/ml) had a significantly ($p < 0.05$) lower number of NK cells than healthy donors ($8.1 \times 10^5$ cells/ml) (Fig. 3).

## MEPB leaves stimulate cytotoxicity and the activation of NK cells

Figure 4 illustrates the apoptotic cell progression in all experiment groups for healthy and breast cancer patient donors. There was a significant ($p < 0.05$) increase of apoptosis in all experiment groups (Fig. 5). In both healthy and breast cancer patient donors, the MDA-MB-231 breast cancer cells did not undergo apoptosis (Group 1; MDA-MB-231 breast cancer cell lines alone). In Group 2, with the addition of MEPB, there was a decrease in living cells, and most of the cells underwent early and late apoptosis, while the percentage of living cells a dropped and apoptotic cell progression was raised in Groups 3 with the

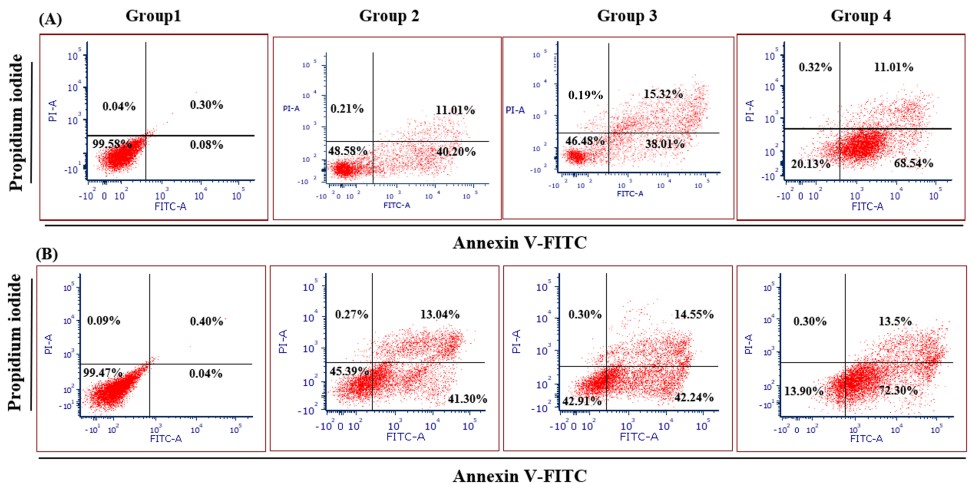

**Figure 4 Apoptosis induction of MDA-MB-231 breast cancer cells by NK cells cytotoxicity.** (A) Healthy donors-experimental groups, (B) breast cancer patients-experimental groups. The data represents the percentage of living, apoptotic, and necrotic cells, respectively.

addition of NK cells, and then further raised in Group 4 with the combination of NK + MEPB. Flow-cytometric analysis showed a commensurate and substantial decrease in the proportion of living cells and an increase in early and late apoptotic activity in Group 4 (MDA-MB-231 breast cancer cell lines+NK cells +MEPB) when compared to Groups 1 (MDA-MB-231 breast cancer cell lines alone), Group 2 (MDA-MB-231 breast cancer cell lines + MEPB), and Group 3 (MDA-MB-231 breast cancer cell lines + NK cells) in healthy and breast cancer patient donors (Fig. 5).

The data obtained from experimental groups of healthy and breast cancer patient donors indicated there was no obvious expression of perforin, granzyme B, and IFN-$\gamma$ of NK cells in Group 1 (MDA-MB-231 breast cancer cell lines alone) and Group 2 (MDA-MB-231 breast cancer cell lines + MEPB); in contrast, there was an increased expression in Group 3 (MDA-MB-231 breast cancer cell lines + NK cells) and Group 4 (MDA-MB-231 breast cancer cell lines+NK cells+ MEPB). The expression of IFN-$\gamma$, perforin, and granzyme B in Group 4 higher than Group 3 throughout the incubation time. There were significant differences among groups ($p <$0.05). On the other hand, the level of perforin, granzyme B, and IFN-$\gamma$ in breast cancer patients-experimental groups were increased compared to that of healthy experimental groups (Fig. 6).

## DISCUSSION

One of the innovative strategies for treating breast cancer is immunotherapy. Innate immune cells are the primary line of defence in the human body against cancer. For instance, NK cells, also known as large innate lymphoid cells, are crucial in natural cancer immune response (*Barrow et al., 2018*). Preclinical studies revealed that NK cells trigger apoptosis against cancer cells and cancer patients has lower NK cell activity, which resulted in cancers' immune escape and their progression (*Burger et al., 2019*; *Chiossone et al.,*

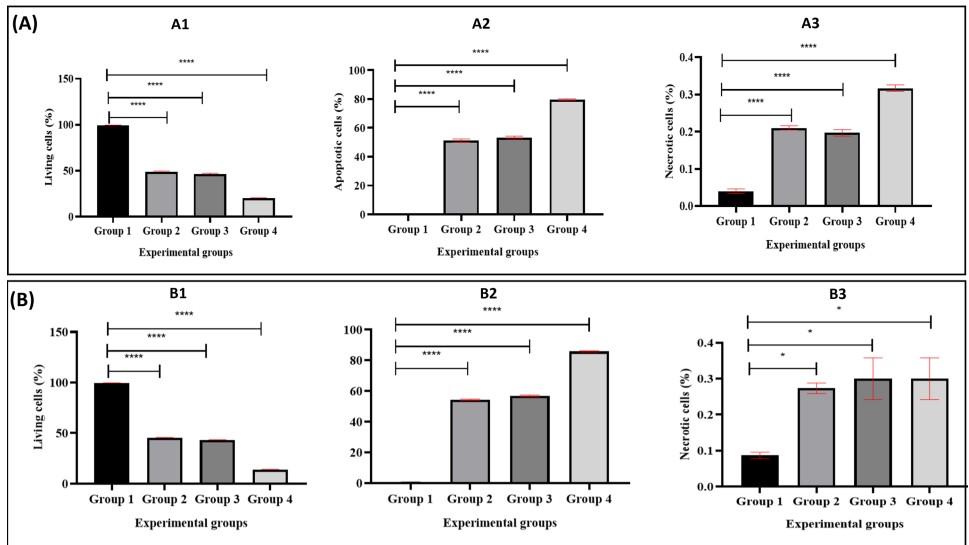

**Figure 5** **The proportion of apoptotic progression of MDA-MB-231 breast cancer cells in healthy donor-experimental groups (A) and breast cancer patient donor-experimental groups (B).** A1 and B1 Represent the percentage of living, A2, B2: apoptotic, and A3, B3: necrotic. The data was expressed as mean $\pm$ SEM. **** $P < 0.0001$ and * $p < 0.01$, significant difference between the experimental groups, $n = 3$.

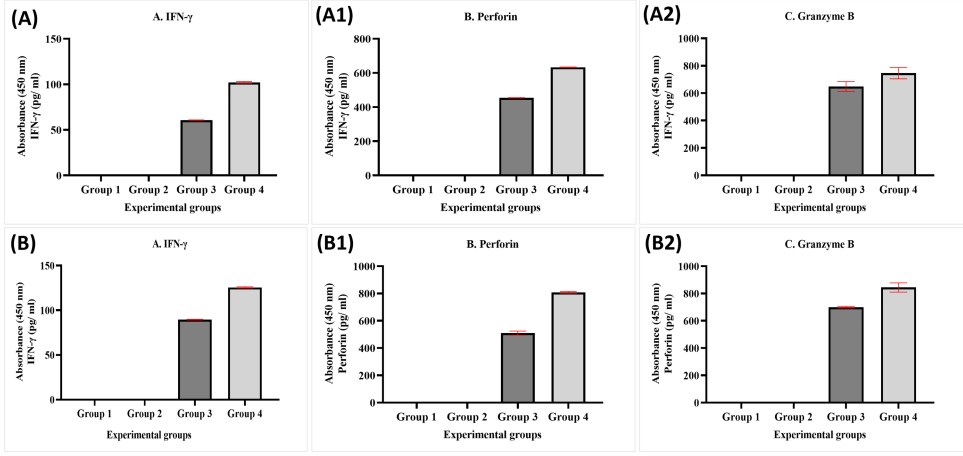

**Figure 6** **ELISA data for IFN- $\gamma$ (A, B), perforin (A1, B1), and granzyme B (A2, B2) levels in both healthy donors ($n = 3$) and breast cancer patients ($n = 3$) for experiment groups.** The data was reported as mean $\pm$ SEM with significant differences ($p < 0.05$).

*2018*). *P. bleo* leaves are abundant in phytochemicals and have been reported in methanol extract based on *Mohd-Salleh et al. (2020a)*. In an animal model, study demonstrated that MEPB leaves did not exhibit toxicity and teratogenic effects. For instance, all animal groups had oestrous cycle regularity, normal body weight, no physical and behavioral signs of toxicity, and normal visceral organ weights. The histological examination also showed no

substantial changes in the visceral organs of all the rat dams. Moreover, the plant extract did not affect the pregnancy outcomes, such as the numbers of corpora lutea and implantation sites, percentages of pre- and post-implantation death, gravid uterine weight, number of live foetuses, foetal body weight, and foetal sex ratio. No outward deformities were seen in any of the experimental groups, as determined by gross examination of the fetuses (*Khalaf et al., 2024*).

NK cell activation has been used in various clinical settings to kill cancer cells. Innovative treatments for NK cells include cytokines, allogeneic NK cells, autologous NK cells, and gene-edited CAR-NK cell immune therapy (*Chu et al., 2022*). A clinical trial published in 2019 detailed the impact of autologous NK cell treatment given to patients with advanced lung cancer. The NK cell therapy was administered to 13 patients without any further treatment. Two patients (15.4%) were assessed as having a progressing condition after three months, while eleven patients (84.6%) had a stable disease (*Xie et al., 2019*). A prior study found that haploidentical allogeneic NK cells may be safely transplanted to metastatic melanoma and renal cell cancer patients (*Miller et al., 2005*). Furthermore, another previous study has published the results of a phase I first-in-human clinical trial using CD33-CAR NK cells for relapsed or resistant acute myeloid leukemia patients. Three individuals were enrolled and their safety was evaluated; however, no untoward occurrences were noted (*Tang et al., 2018*). Trial carried out on rats administered with curcumin demonstrated that the higher dosage of curcumin caused NK cells to produce more nitric oxide, which in turn caused apoptosis in AK-5 and YAC-1 cancer cells (*Bhaumik, Jyothi & Khar, 2000*). Previous study reported that curcumin can triggering NK cells to induce apoptosis against K562 cancer cells in curcumin-treated rats (*Yadav et al., 2005*). A study conducted by Lu and Chen found that resveratrol showed the ability to elevated NK92 cells cytotoxicity on K562, HepG2, and A549 cancer cells *via* elevated JNK and ERK1/2 MAP kinase activity, expression of perforin, NKG2D receptor expression (*Lu & Chen, 2010*).

Medicinal plants exhibit various biological benefits, such as immunomodulatory, anti-oxidant, anti-bacterial, anti-inflammatory, and anti-cancer properties (*Seca & Pinto, 2018*; *Zebeaman et al., 2023*). According to *in vitro* and *in vivo* investigations, medicinal plants have been proven to alter several cytokines (*Deplanque, Shabafrouz & Obeid, 2017*). Evidence suggests that cytokines have an essential role in the cancer immune response (*Taouk et al., 2019*).

IFN-$\gamma$ is categorised as a pro-inflammatory cytokine and is the primary effector of cellular immunity. T cells, NK cells, B cells, and neutrophils, are responsible for releasing IFN-$\gamma$ in the human body. IFN-$\gamma$ acts as antiviral, antibacterial, and anti-cancer (*Chiossone et al., 2018*). IL-12 is known as a pro-inflammatory cytokine, which produced by antigen presenting cells, including dendritic cells and macrophages (*Baggio et al., 2017*). Previous investigation showed that IL-12 potently boosts NK cells' cytotoxicity (*Deplanque, Shabafrouz & Obeid, 2017*), increase the ADCC pathway (*Campbell et al., 2015*), and MHC class I expression (*Suzuki et al., 1998*) in cancer cells to facilitate the destruction. On the other hand, IL-18 is produced by immune and non-immune cells, including macrophages, dendritic cells, monocytes, T cells, B cells, Langerhans cells, endothelial cells, chondrocytes, osteoblasts, and intestinal epithelial cells (*Yasuda, Nakanishi & Tsutsui, 2019*). Studies have

confirmed that IL-18 has an essential role in boosting the cytotoxic effects of NK cells by upregulating the production of IFN-$\gamma$ (*He, Hara & Núñez, 2016*) and perforin (*Hyodo et al., 1999*). Perforin and granzyme B protein play a role in the immune response to defence against virus infection and cancer. Both are expressed primarily by NK cells and cytotoxic T lymphocytes (CTLs) (*Baggio et al., 2017*). In the current investigation, IFN-$\gamma$, IL-12, IL-18, perforin, and granzyme B expression were determined in the supernatant of healthy blood samples by ELISA. MEPB leaves were found to activate immune cells to express IFN-$\gamma$, IL-12, IL-18, perforin, and granzyme B in blood samples after 20 hrs of treatment. According to a study conducted *in vitro*, ELISA data demonstrated that *Ocimum sanctum* increased the expression of IFN-$\gamma$ in the supernatant of blood donors that were incubated in RPMI-1640 medium; stimulated with PHA at 5 $\mu$g/ml and *Escherichia coli*-derived lipopolysaccharide (LPS) at 25 $\mu$g/ml (*Mondal et al., 2011*). In a rat model, ELISA results showed that seed extract of Black cumin extract increased IFN-$\gamma$ release by lymphocyte (*Akrom et al., 2019*). Another *in vitro* research found that RAW 264.7 cells treated with polyphenolic polysaccharide-proteins extracted from *Echinacea purpurea* and *Erigeron canadensis* resulted in higher levels of IL-12, as revealed by ELISA (*Paulovičová et al., 2022*). In a murine model, ELISA results revealed that *Berberis vulgaris* extract increased the level of IL-12 in female mouse splenocytes (*Aziz et al., 2015*). On the other hand, it was shown in another study that the oral treatment of Phellinus igniarius extract at doses of 250 and 500 mg/kg significantly elevated IL-18 levels in mice implanted with Sarcoma 180 cancer cells (*Chen et al., 2011*). The increase of perforin and granzyme B was also found in rats subjected to chronic stress and then treated with *Hippophae rhamnoides* L. oil (*Diandong et al., 2016*).

IL-8 is a pro-inflammatory cytokine generated by neutrophils, macrophages, and cancer cells (*Hidaka et al., 2002*). It plays a crucial part in controlling inflammatory reactions by attracting immune cells to the infection site and promoting phagocytosis. Studies have documented that cancer cells have been shown to release IL-8 to promote their own development or block apoptosis (*Liu et al., 2016*). While IL-10 is an anti-inflammatory cytokine released by macrophages, eosinophils, mast cells, dendritic cells, NK cells, and cancer cells, acts as an immunosuppressive factor and has pro-tumour and anti-cancer activity (*Oft, 2014*). IL-10 has a potent role in reducing the production of IFN-$\gamma$, IL-12, and IL-18 *via* suppressing immune cells. We analysed the expression of IL-8 and IL-10 in supernatant of healthy donors' blood by ELISA method. Our data indicated that MEPB leaves could reduce the expression of IL-8 and IL-10 after 20 hrs of treatment in blood samples. Other natural product, curcumin downregulated IL-8 production at doses 50 and 100 $\mu$M in SUIT-2 cancer cells (*Hidaka et al., 2002*). In vitro research finding that *Origanum syriacum*, *Chrysanthemum coronarium*, *Rosmarinus officinalis*, *Inula viscosa*, *Arbutus andrachne*, and *Punica granatum* extracts can decrease IL-10 in PBMCs supernatants of healthy donors' blood (*Assaf et al., 2016*).

Fresh blood samples from healthy donors and cancer patients were used to isolate NK cells in this experiment. Two markers were used in flow-cytometry to detect human NK cells, including neural cell adhesion molecule CD56 (NCAM) expression and T cell marker CD3 expression, as the human NK cell phenotype is defined by the presence of CD56 and
a lack of CD3 (*Chiossone et al., 2018*). Our NK cells' purity was 87.09% and the average purity of human NK cells recovered from PBMCs using the human NK cell isolation kit was 86%. The current data reported that the number of NK cells in cancer patients (4.3 $\times\ 10^5$ cells/ml) was lower than in healthy donors (8.1 $\times\ 10^5$ cells/ml). A low NK cell count is commonly seen in cancer patients due to impaired NK cell homeostasis, including disrupted cell cycle phases and maturation (*Chiossone et al., 2018*).

Previous recorded research has shown that medicinal plants can induce apoptosis in cancer cells (*Lucas et al., 2010*). This is achieved by activators of the immune system, including NK cells (*Azadmehr et al., 2016*). NK cell activation can trigger apoptosis in cancer cells by several pathways, including the release of cytotoxic granules containing pore-forming perforin protein and effector serine proteases such as granzyme B (*Baggio et al., 2017*). Perforin creates a pore in the cell membrane, allowing granzyme B to enter and trigger the destruction or programmed cell death of the target cells. Furthermore, NK cell-mediated lysis of cancer cells is reflected by IFN-$\gamma$ release (*Chiossone et al., 2018*). IFN-$\gamma$ can suppress cell proliferation, downregulate angiogenesis, and upregulate MHC class I expression in the tumour microenvironment. Based on previous investigations, IFN-$\gamma$ singling can block or moderate the progression of cancer cells by cell cycle arrest (G0/G1) (*Akrom et al., 2019*). Besides, it can also reduce the progression of breast cancer cells by raising the expression of cell cycle inhibitor proteins (p27Kip, p16, or p21), leading to apoptosis (*Kochupurakkal et al., 2015*). A study investigated by *Mohd-Salleh, Wan-Ibrahim & Ismail (2020b)* reported that *P. bleo* leaves extract exhibited alteration in cell cycle progression. According to their data obtained from flow-cytometry showed elevated Hela cancer cells in G0/G1 phase and decreased in S and G2 phase after treated with ethyl acetate extract of P. bleo leaves.

We examined the effects of MEPB on NK cells-IFN-$\gamma$, perforin, and granzyme B cytotoxicity against MDA-MB-231 breast cancer cells by ELISA method and apoptosis assay. In both experimental groups, the current data obtained from apoptosis assay revealed that there were no apoptotic cells in Group 1 (MDA-MB-231 breast cancer cell lines alone), while decrease in the percentage of living cells and increased in early and late apoptosis in Group 2 (MDA-MB-231 breast cancer cell lines + MEPB leaves). The current data indicated that NK combination with MEPB can increase apoptotic levels (early and late apoptosis) and decrease living cells against MDA-MB-231 breast cancer cells in Group 4 compared with Group 3 (MDA-MB-231 breast cancer cell lines + NK cells). ELISA data exhibited there were expression of IFN-$\gamma$, perforin, and granzyme B in Group 1 (MDA-MB-231 breast cancer cell lines alone) and Group 2 MDA-MB-231 breast cancer cell lines + MEPB), while there was expression of IFN-$\gamma$, perforin, and granzyme B in Group 3 (MDA-MB-231 breast cancer cell lines + NK cells). The level of IFN-$\gamma$, perforin, and granzyme B by NK cells were significantly elevated after being treated with MEPB (Group 4), leading to increase apoptosis on MDA-MB-231 breast cancer cells. Hou and co-workers found that flavonoids derived from *Hippophae rhamnoides* had the ability to increase the cytotoxicity of NK cells against K562 cancer cells by perforin/granzyme B production (*Hou et al., 2017*). In the mouse model, a lower concentration (50–200 µg/ml) of the extract from the *Scrophularia variegate* plant showed the ability to raise NK cells

cytotoxicity by expression of IFN-$\gamma$ against Yac-1 cancer cells, according to ELISA data (*Azadmehr et al., 2016*). A study conducted on ginseng cambial meristematic cells-treated mice showed that methanol extract of ginseng resulted in increased NK cells activity and granzyme B in cambial meristematic cells (*Jang et al., 2015*). A prior study reported that oral administered of aqueous extract of *Panax ginseng* revealed elevated NK cells cytotoxicity on wild-type (WT) C%7BL/6 (WT B6) and BALB/c mice *via* IFN-$\gamma$-dependent manner (*Takeda & Okumura, 2015*). Furthermore, previous study documented that Apigenin and luteolin (12.5 μg/ml and 25 μg/ml) exhibited ability to raised NK cells cytotoxicity on lung cancer cells *via* elevated expression of perforin and granulysin (*Oo et al., 2021*).

### Study limitations

Although, the current study has thoroughly investigated the plant leaves and NK cells, the *in vivo* clinical trials on the humans could provide deeper understanding of the exact mechanism.

## CONCLUSIONS

Conclusively, *Pereskia bleo* leaves could potentially modulate the anti-cancer immune response by up-regulating IFN-$\gamma$, IL-12, IL-18, perforin, and granzyme B, while downregulating IL-8 and IL-10 in healthy blood samples. Specifically, our findings reported that MEPB has the ability to stimulate the activation and cytotoxicity of NK cells towards the triple-negative human breast cancer cell lines (MDA-MB-231) by increasing the NK cells-activating cytokines IFN-$\gamma$, perforin, and granzyme B, three important chemicals that aid in the death of cancer cells. The results shows that *P. bleo* leaves have a considerable capacity to increase cytokine synthesis, which is essential for immunological regulation. Thus, the results of current study might help researchers better understand the role of *P. bleo* leaves in apoptosis induction and cytotoxicity of NK cells against cancer cells. Further studies are needed to see the possibility of the medicinal plants to be used as immunotherapeutic approaches, helping to create new medications that better target cancer by utilizing the body's immune system. Further potential avenues for future research could include determining the active chemicals responsible for these benefits and delving deeper into the precise mechanisms by which these plants regulate immune responses.

### Funding

This study was supported by the Fundamental Research Grant Scheme (FRGS/1/2019/WAB11/USM/03/1) from the Ministry of Higher Education Malaysia. The funders had no role in study design, data collection and analysis, decision to publish, or preparation of the manuscript.

### Grant Disclosures

The following grant information was disclosed by the authors:

the Fundamental Research Grant Scheme: FRGS/1/2019/WAB11/USM/03/1.

## Competing Interests

The authors declare there are no competing interests.

## Author Contributions

- Taif Kareem Khalaf conceived and designed the experiments, performed the experiments, prepared figures and/or tables, authored or reviewed drafts of the article, and approved the final draft.
- Norzila Ismail conceived and designed the experiments, performed the experiments, authored or reviewed drafts of the article, and approved the final draft.
- Nor Amalia Nazri conceived and designed the experiments, performed the experiments, authored or reviewed drafts of the article, and approved the final draft.
- Naveed Ahmed analyzed the data, prepared figures and/or tables, authored or reviewed drafts of the article, and approved the final draft.
- Aidy Irman Yajid performed the experiments, authored or reviewed drafts of the article, and approved the final draft.
- Rohimah Mohamud conceived and designed the experiments, analyzed the data, authored or reviewed drafts of the article, and approved the final draft.
- Ramlah Kadir analyzed the data, authored or reviewed drafts of the article, and approved the final draft.

## Human Ethics

The following information was supplied relating to ethical approvals (*i.e.*, approving body and any reference numbers):

Ethical approval was obtained from JEPeM, Universiti Sains Malaysia, with the approval code: USM/JEPeM/20080430.

## Data Availability

The raw data is available in the Supplemental File.

## Supplemental Information

Supplemental information for this article can be found online at http://dx.doi.org/10.7717/peerj.18420#supplemental-information.

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
