# Peer review of "Pereskia bleo augments NK cell cytotoxicity against triple-negative breast cancer cells (MDA-MB-231)"

_PeerJ, doi:10.7717/peerj.18420_

## Round 0.1 · original submission · Minor Revisions

The manuscript under review is well designed and the article will be of interest for the journal's readership. The authors are requested to address the reviewers' comments to enhance the overall readability of the manuscript. It also recommended to have a native English speaker review the article for grammatical errors.

Reviewer 1 ·

Basic reporting

For many of the figures and some of the text, it can be confusing to refer to the groups without their descriptions. It would be helpful to add labels to each group in the figure (example: Group A, + PHA +MEPB leaves), and in the text this would be helpful too (for example: In Group 2, with the addition of MEPB leaves, there was a decrease in living cells, and most of the cells underwent early and late apoptosis, while the percentage of living cells a dropped and apoptotic cell progression was raised in Group 3 with the addition of NK cells, and then further raised in Group 4 with the combination of NK + MEPB leaves.). Figure 2 and 3 need better labels for the axes, to define what was being stained and not just the fluorophore. In Figure 2, the percentages for *each* replicate should be shown and quantified. The p values should be reported in the figure for each comparison. The information and conclusion from table 1 is not clear. Needs some overall grammatical editing.

Experimental design

For the blood, how do you know if the cytokines are being produced by NK cells? Did you test IL-8 and IL-10 in the in vitro coculture experiments? Why does group 2B seem to have more cell death than 2A when these should be identical? It would be nice to have graphs 4+5 together to compare samples from breast cancer patients to healthy patients.

Validity of the findings

The findings are interesting and conclusions are well stated.

·

Basic reporting

The current manuscript by Ismail and colleagues reports on the previously unexplored property of methanol extract of P. bleo leaves (MEPB) in enhancing natural killer (NK) cell cytotoxicity against a breast cancer cell line. The workflow includes a straightforward cytokine assay followed by a cellular apoptosis assay using basic ELISA and flow cytometry techniques. The simple yet effective experimental design aligns well with the proposed research question and provides supporting data for the reported properties of MEPB.
The authors employ clear professional English language and maintain a well-structured flow of information, beginning with a review of previous studies and the existing knowledge gap, followed by a description of the current research that showcases the NK cell-mediated anticancer properties of MEPB leaves against breast cancer. This research not only contributes a new discovery to the field of P. bleo's anticancer properties but also demonstrates potential for using these leaves in combination therapy for treating breast cancer, and possibly other types of cancer as well.
Although the current manuscript is strong enough for acceptance, I propose a few modifications to more clearly portray the results and make the communication more accessible to audiences from different fields. For example, figures throughout the manuscript are well-organized and labeled; however, the flow cytometry density plot in Figure 3 should be labeled with the percentage of cell populations in each quadrant for the apoptosis assay results.

Experimental design

1. I commend the authors for their well-structured experimental design and thoughtful planning in addressing the proposed research question. In addition, the manuscript is written in clear, professional language with a well-organized flow of results and their corresponding discussion. However, I recommend providing more detailed information in the methods section. For example:
a) Line 121: “PHA was used as… the samples.” Please specify the concentration of PHA used in this experiment, as well as the solvent in which the stock solution was prepared.
b) Line 126: “The experiment was achieved…ELISA kit from” Please provide brief details about the ELISA procedure, including: (i) the amount of blood supernatant used, (ii) the antibody dilution applied in the assay, (iii) incubation time points, and any other specifics that contributed to achieving optimal results.
c) Line 155: MDA-MB-231 breast cancer cell lines were co-cultured at a 20:1 ratio. Was an alternative ratio tested, or was this ratio chosen based on a reported protocol? If the latter, a reference is recommended to include.
2. The method section for the apoptosis assay is well described (Line 162-170); however, I recommend including additional details about the flow cytometry experiment, such as the cell count used for each experiment and the method used for compensation.
3. Although the research question is well-defined, focusing on the ability of MEPB to stimulate NK cell-mediated cytotoxicity, there is a lack of context regarding whether other reports exist on P. bleo leaves activating other immune cells.
4. Figure 1: The flow cytometry density plot should include the percentage of the cell population in each quadrant. While the current figure demonstrates the purity of the isolated NK cells, it is recommended (although not mandatory for this publication) to include both a negative and a positive control to rule out the possibility of nonspecific antibody binding on the cell surface. Isotype control antibody serves as great negative control in this case.
5. Tables 3 and 4 are highly informative for readers to understand the ELISA results for the cytokine assay. However, a visual representation, such as a bar plot, is recommended to convey the results more effectively.
6. Line 97: “our previous experiment...towards breast cancer.” Please include the reference of the previous report. It is requested to mention this in details about whether the cytotoxicity was checked with the apoptosis assay of any other assay like MTT assay.

Validity of the findings

I appreciate the author’s inclusion of robust statistical calculation in each experiment, discussion to support the research. However, in the discussion section, it is recommended
a) A more comprehensive discussion is needed to connect the previously reported apoptotic properties (via cell cycle arrest mechanism) of P. bleo leaves (Nutrition and cancer, 2020, 72.5, 826-834) with their currently observed role in augmenting immunostimulants. Specifically, the results described in lines 182-186 require a deeper discussion to clarify how they support the conclusion that MEPBs play a crucial role in immunostimulant activity.
b) Figure 3 illustrates that both Group-2 (A and B) exhibit a significant apoptosis profile, progressing through both early and late apoptosis stages. This effect is attributed to the cytotoxic properties of P. bleo leaves, as reported previously (Nutrition and Cancer, 2020, 72.5, 826-834). In contrast, Groups 3 and 4 show a higher number of apoptotic or necrotic cells, with Group 4 displaying more than Group 3. This difference is likely due to the proposed effects of MEPBs on NK cells. This finding should be clearly addressed in the discussion section with appropriate references.
c) A brief outlook in the conclusion section should highlight how these findings could be leveraged for advancements in cancer diagnostics and drug development. Specifically, the apoptotic and cytotoxic properties observed could be explored to develop novel therapeutic agents or diagnostic tools that target cancer cell death more effectively.

Additional comments

I recommend acceptance with minor revisions. The primary suggestion is to provide more detailed information in the Materials and Methods section to ensure that all necessary details are included for replication of the study.

·

Basic reporting

The article is written in English and uses appropriate scientific terminology. The overall language is professional and courteous. Most technical terms are used correctly. However, there are instances where more precise language could be used. For example, when discussing the effects of MEPB, phrases like "could reduce" or "were found to activate" could be more definitive if the data supports it. The use of "healthy blood donors" and "healthy individuals" is inconsistent throughout the paper. Standardizing this terminology would improve clarity.

The introduction provides a general background on NK cells and their role in cancer immunosurveillance very well. However, the rationale for choosing Pereskia bleo is not well established. While it's mentioned that P. bleo has been used in traditional medicine, there's no information about why this particular plant was selected for study against TNBC. Additionally, an explicit statement of the hypothesis being tested is not present in the introduction. Including this would strengthen the paper's focus.

The paper generally follows a standard scientific article structure with Introduction, Materials and Methods, Results, Discussion, and Conclusion sections. Figures are relevant to the content and are generally well-labeled and described. However, some improvements are needed. The figure legends for figures 1 and 2 are repeated- the same sentence is written twice.

There's no mention of raw data availability or a data sharing statement, which is increasingly important for reproducibility in scientific research. The paper would benefit from a clear statement about where and how other researchers can access the underlying data.

Experimental design

The gap in knowledge that this study aims to fill is not explicitly stated. A clear statement of what is unknown and how this study addresses that gap would improve the introduction.

The sample size (n=3 for both healthy donors and breast cancer patients) is quite small, which limits the statistical power and generalizability of the results.
There's no mention of how the breast cancer patients were selected or at what stage of treatment they were, which could affect NK cell counts and activity.

The study design has not checked for toxicity in non-cancerous cells

In the methods section for the isolation of NK cells, it is not clear how much blood was used. Was it the same for both groups? On a related note, in Figure 2, the number of NK cells is shown but is missing the percentage purity.

Validity of the findings

The study presents new findings on the effects of Pereskia bleo on NK cell activity against triple-negative breast cancer cells. While it builds on existing knowledge about plant-derived immunomodulators, it does offer novel insights. The topic of natural product-based immunotherapy for cancer is of broad interest in the fields of oncology and immunology, not limited to a niche audience. It is worth noting that the authors do report both stimulatory and inhibitory effects of MEPB on different cytokines, showing a willingness to report all results, not just positive ones.

In the discussions, the paper doesn't adequately discuss the limitations of drawing broad conclusions from in vitro studies. The effects observed in cell culture may not translate directly to in vivo situations. The conclusions could be improved by more clearly stating what questions remain unanswered and what future studies are needed to further validate these findings.

Additional comments

No comment

Reviewer 4 ·

Basic reporting

-Please note that manuscripts with inadequate language quality will not be accepted by the journal. Language revisions are necessary for your manuscript.
-Please include more detailed information about NK cell therapy for cancer patients and the combination of natural products with cellular immunotherapy in both the introduction and discussion sections. Additionally, discuss the advantages of the study for future research and address the limitations of the current study.
-In the manuscript, Tables 1 and 2 present the experimental methodology. To enhance reader comprehension, these should be converted into figures.
Figures 1 and 3 should have the y-axis and x-axis labelled, such as CD3 or CD56 or PI and anexinV, and the values for Q1, Q2, Q3, and Q4 should be included as representative data.
-Figures 1 and 3 are incomplete due to the image being cropped.
-Figures 3, 4, and 5 convey similar information. They can be combined into a single figure or separated into distinct figures for healthy donors and breast cancer patients, respectively.

Experimental design

no comment

Validity of the findings

This study demonstrates that P. bleo enhances NK cell activation, leading to increased lysis of triple-negative breast cancer cells. The findings suggest that MEPB leaves could serve as a valuable immunostimulant in cancer therapy. The novelty of this research lies in its exploration of P. bleo as a natural compound to boost NK cell activity against a challenging cancer type, providing a new avenue for immunotherapeutic strategies.

Additional comments

-The cytotoxicity of P. bleo against MDA-MB-231 cells and its effects on NK cells should be demonstrated.
The phytochemical profile of the methanol extract of P. bleo should be investigated to identify the major constituents responsible for activating NK cell function and sensitizing cancer cells. These findings should provide further insights for discussion.
-The safety of the extracts should be evaluated and investigated to ensure their suitability for future use in patients.

---

## Round 0.2 · Minor Revisions

The revised manuscript has addressed most of the reviewer comments satisfactorily. I recommend minor revision based on the comments. As one of the reviewers mentioned, it is crucial to speculate and highlight whether the cytokines were released from other cell types if they weren't verified that NK cells were responsible. .

Reviewer 1 ·

Basic reporting

Overall the authors have worked to address the comments. However, a few comments need clarification. The replicates in Fig 2 are not shown, nor is the p value. Table 1 needs more clarification- it is not clear that the ug/mL refers to MEPB leaves. Lastly, if the authors did not verify that the cytokines in the blood are from NK cells, they need to speculate in the text that these cytokines may come from other cell types.

Experimental design

This is sufficient and well thought out.

Validity of the findings

Interesting findings from the authors.

·

Basic reporting

The authors have successfully addressed all the concerns raised in the review.

Experimental design

The authors have successfully addressed all the concerns raised in the review.

Validity of the findings

The authors have successfully addressed all the concerns raised in the review.

·

Basic reporting

Upon revision, the authors have addressed all comments provided.

Experimental design

Upon revision, the authors have addressed all comments provided.

Validity of the findings

Upon revision, the authors have addressed all comments provided.

---

## Round 0.3 · accepted · Accept

All the reviewer comments have been addressed to satisfaction.